# Eight Weeks Unsupervised Pulmonary Rehabilitation in Previously Hospitalized of SARS-CoV-2 Infection

**DOI:** 10.3390/jpm11080806

**Published:** 2021-08-18

**Authors:** Vasileios T. Stavrou, Konstantinos N. Tourlakopoulos, George D. Vavougios, Eirini Papayianni, Katerina Kiribesi, Stavros Maggoutas, Konstantinos Nikolaidis, Evangelos C. Fradelos, Ilias Dimeas, Zoe Daniil, Konstantinos I. Gourgoulianis, Stylianos Boutlas

**Affiliations:** 1Laboratory of Cardio-Pulmonary Testing and Pulmonary Rehabilitation, Department of Respiratory Medicine, Faculty of Medicine, University of Thessaly, 41110 Larissa, Greece; kostisst@hotmail.com (K.N.T.); dantevavougios@hotmail.com (G.D.V.); eirinipapayianni@gmail.com (E.P.); katerinakir08@hotmail.com (K.K.); smagkoutas@uth.gr (S.M.); kon.nikolaidis93@gmail.com (K.N.); zdaniil@uth.gr (Z.D.); kgourg@uth.gr (K.I.G.); 2Department of Respiratory Medicine, Faculty of Medicine, University of Thessaly, 41110 Larissa, Greece; dimel13@hotmail.com (I.D.); sboutlas@gmail.com (S.B.); 3Scientific Research Associate, Department of Neurology, Athens Naval Hospital, 11521 Athens, Greece; 4Nursing Department, University of Thessaly, 41500 Larisa, Greece; evagelosfradelos@hotmail.com

**Keywords:** pulmonary rehabilitation, unsupervised exercise, post COVID-19

## Abstract

The aim of our study was to determine the impact of unsupervised Pulmonary Rehabilitation (uns-PR) on patients recovering from COVID-19, and determine its anthropometric, biological, demographic and fitness correlates. All patients (*n* = 20, age: 64.1 ± 9.9 years, 75% male) participated in unsupervised Pulmonary Rehabilitation program for eight weeks. We recorded anthropometric characteristics, pulmonary function parameters, while we performed 6 min walk test (6 MWT) and blood sampling for oxidative stress measurement before and after uns-PR. We observed differences before and after uns-PR during 6 MWT in hemodynamic parameters [systolic blood pressure in resting (138.7 ± 16.3 vs. 128.8 ± 8.6 mmHg, *p* = 0.005) and end of test (159.8 ± 13.5 vs. 152.0 ± 12.2 mmHg, *p* = 0.025), heart rate (5th min: 111.6 ± 16.9 vs. 105.4 ± 15.9 bpm, *p* = 0.049 and 6th min: 112.5 ± 18.3 vs. 106.9 ± 17.9 bpm, *p* = 0.039)], in oxygen saturation (4th min: 94.6 ± 2.9 vs. 95.8 ± 3.2%, *p* = 0.013 and 1st min of recovery: 97.8 ± 0.9 vs. 97.3 ± 0.9%), in dyspnea at the end of 6 MWT (1.3 ± 1.5 vs. 0.6 ± 0.9 score, *p* = 0.005), in distance (433.8 ± 102.2 vs. 519.2 ± 95.4 m, *p* < 0.001), in estimated O_2_ uptake (14.9 ± 2.4 vs. 16.9 ± 2.2 mL/min/kg, *p* < 0.001) in 30 s sit to stand (11.4 ± 3.2 vs. 14.1 ± 2.7 repetitions, *p* < 0.001)] Moreover, in plasma antioxidant capacity (2528.3 ± 303.2 vs. 2864.7 ± 574.8 U.cor., *p* = 0.027), in body composition parameters [body fat (32.2 ± 9.4 vs. 29.5 ± 8.2%, *p* = 0.003), visceral fat (14.0 ± 4.4 vs. 13.3 ± 4.2 score, *p* = 0.021), neck circumference (39.9 ± 3.4 vs. 37.8 ± 4.2 cm, *p* = 0.006) and muscle mass (30.1 ± 4.6 vs. 34.6 ± 7.4 kg, *p* = 0.030)] and sleep quality (6.7 ± 3.9 vs. 5.6 ± 3.3 score, *p* = 0.036) we observed differences before and after uns-PR. Our findings support the implementation of unsupervised pulmonary rehabilitation programs in patients following COVID-19 recovery, targeting the improvement of many aspects of long COVID-19 syndrome.

## 1. Introduction

The severe acute respiratory syndrome coronavirus 2 (SARS-CoV-2) is the causative agent for COVID-19. The COVID-19 pandemic, in turn, is responsible for an unprecedented rise in morbidity and mortality globally [1]. A growing body of evidence suggests that COVID-19 inflicts long-lasting consequences in survivors [2]. These residual, long-term effects of SARS-CoV-2 infection include fatigue, dyspnea, chest pain, cognitive disturbances, arthralgia, and an overall decline in quality of life. Collectively, they characterize the post-acute COVID-19 syndrome (PASC) [3,4,5]. There is a wide spectrum of PASC presentations. These include breathlessness, dysfunctional breathing, oxygen requirements, post-viral cough, cardiovascular muscular changes, anxiety, post-traumatic stress disorders, sleep disorders, chronic fatigue, cognitive impairment, and sarcopenia [6]. PASC treatment requires access to a full, multidisciplinary rehabilitation service in order to ameliorate these symptoms [7]. Pulmonary rehabilitation (PR) represents a comprehensive approach towards the care and improvement of the functional status of patients with respiratory diseases [8]. PR reduces symptom burden, increases functional ability, and improves quality of life. PR programs can be delivered within the hospital setting, outpatient, home-based, and/or even remotely supervised [9]. Telerehabilitation in particular has been utilized in addressing the needs of COVID-19 survivors [10]. The average duration of international PR programs ranges from 6-to-9 weeks, with some providing ongoing maintenance programs. Exercise training is considered the foundation of PR, comprising 76–100% of programs internationally [11]. There is insufficient information in the literature about the impact of unsupervised telerehabilitation in aspects such as fitness indicators, oxidative stress markers, strength, stamina, body composition and quality of sleep [12].

The aim of our study was to determine the impact of unsupervised PR (uns-PR) on patients recovering from COVID-19 and determine its anthropometric, biological, demographic and fitness correlates.

## 2. Materials and Methods

### 2.1. Study Population

Twenty consecutive COVID-19 survivors, previously hospitalised in the University Hospital of Larissa, Greece (Figure 1) from September 2020 to December 2020 (second wave of COVID-19 in Greece) were included in our study (Table 1). All patients were selected two months after discharge from hospital.

Inclusion criteria were:patients no longer require oxygen,their fever has resolved for a consecutive 48-h period without any medication to reduce their fever orstable patients even though they still require supplemental oxygen supposing oximetry self-monitoring due to limited hospital resources according to NIH discharge criteria for COVID-19 patients and Hellenic guidance for COVID-19 pneumonia diagnosis.

Exclusion criteria were:the absolute and relative contraindications for 6 MWT [13],balance test stork stand with open eyes < 30 s [14],BMI ≥ 40 kg/m^2^,mental illness andany form of musculoskeletal disability which could impair maximum exercise capacity [15,16].

All patients participated in the unsupervised Pulmonary Rehabilitation program for 8 weeks.

In our study, we did not elect to use a control group for several reasons. First and foremost, a control group of COVID-19 survivors not undergoing rehabilitation would be unethical. Secondly, comparisons with other pulmonary disease groups would fail to produce meaningful associations, these diseases are distinctly characterized by unique features and clinical course.

#### Study Ethics

The study was approved by the Institutional Review Board (IRB)/Ethics Committee (EC) of the University Hospital of Larissa (IRB/EC approval reference number: 15314/21-04-2021). All participants had provided written informed consent, in accordance with the Helsinki declaration [17] and personal data according to European Parliament and of the Council of the European Union [18].

### 2.2. Measurements

Patients included in our study provided demographics and complete medical history.

*Anthropometric characteristics and body composition*: Anthropometric measurements included body height (Seca 700, Hamburg, Germany) and chest circumference difference between maximal inhalation and exhalation (Δchest, Seca 201, Hamburg, Germany). The chest circumference was measured in the upright position, after the abduction of the upper limbs and between the 6th and 7th ribs [19]. Neck circumference was measured between the 3rd and 4th vertebrae, waist circumference and calculated the waist-hip ratio [20] (Seca 201, Hamburg, Germany) as well as circumference differences between right and left arm (Δarm), thigh (Δthigh), and calf (Δcalf). All measurements were made three times in order to ensure consistency. Body mass, body composition, and total body water (Tanita MC-980, Arlington Heights, IL, USA) measurements were also included. Other relevant measurements and derived patient metadata included body mass index, BMI = [Weight(kg)/Height^2^(m)] and body surface area, BSA = [(Height(cm) × Weight(kg))/3600] [21].

*Pulmonary function test*: All participants underwent standard spirometry and lung volume measurements, in line with ATS/ERS guidelines [22]. Maximal flow-volume loops were conducted for each subject with sitting position using MasterScreen-CPX pneumotachograph (VIASYS HealthCare, Germany). For each pulmonary function test, three maximal flow-volume loops were obtained to determine FVC and FEV_1_; the largest one was retained to calculate the ratio of FEV_1_ to FVC (FEV_1_/FVC). The diffusing capacity for carbon monoxide (DLCO) was recorded according to ATS/ERS guidelines [22].

*Oxidative stress biomarkers*: A 10 mL sample of peripheral venous blood was drawn from each patient at 10.00 a.m., 20 min prior to the 6 MWT. The patient had fasted the previous night. Subsequently, the sample was used to measure reactive oxygen metabolites’ levels (d-ROMs test) and the plasma antioxidant capacity (PAT test) (Free Radical Analytical System, FRAS5, Parma, Italy). The combination of an oxidative metabolite and an estimate of antioxidants allows the evaluation of redox homeostasis in the given sample [23].

*6 MWT*: The 6 Minute walk test (6 MWT) was used in order to assess functional capacity, as described elsewhere [24]. Specific measurements includedO_2_ saturation (SpO_2_) and heart rate (HR) evaluation (Nonin 9590 Onyx Vantage, USA) m at baseline, every 1 min of test, and at the 1st min of recovery [23]. Blood pressure (BP, Mac, Japan) and self-assessed lower extremity fatigue and dyspnea were captured via the Borg Scale CR10 [25]. These measurements were recorded at baseline, end of the test, and at the 1st min of recovery. The total distance and estimated peak O_2_ uptake [26] [O_2_ uptake _(mL/min/kg)_ = 4.948 + 0.023 × distance _(m)_] and metabolic equivalent [METs = O_2_ uptake _(mL/min/kg)_/3.5] were also evaluated as measures of functional capacity.

*Pulmonary rehabilitation exercise program*: The training program lasted 8 weeks (Table 2), while each patient took part in 3 training sessions per week. The duration of each training session was about 100 min. Each training session included a (i) warm-up (5 min) and (ii) recovery set (5 min) with flexibility and mobility exercises, (iii) the aerobic exercise set with walking (50 min), (iv) the set with yoga exercises for breathing and/or proprioception (20 min) and (v) the set with multi-joint strength exercises (20 min). In the aerobic exercise set, the patients performed walking on a flat and hard surface and every five minutes patients checked their heart rate and oxygen saturation; and subsequently recorded the total distance covered was recorded.

Adherence to the program was determined via 2 phone calls per week prior to the visit. Each call focused on whether the patients were able to follow the instructions, perform them on a daily basis, and troubleshooting.

*Nutrition recommendations*: Personalized diet recommendations focused on food consumption, conservation, and body weight loss. They were issued according to body composition (percent of body fat, visceral fat score, muscle mass, etc.) and resting metabolic rate (RMR). The nutrition programs were formed in accordance with the Mediterranean Diet [27].

*Sleep quality assessment*: Prior to 6 MWT all participants answered Pittsburgh Sleep Quality Index (PSQI) questionnaire [28].

All sessions were performed in the Laboratory of Cardio-Pulmonary Testing and Pulmonary Rehabilitation (University of Thessaly), with the environmental temperature at 23.2 ± 3.1 °C and humidity 34.6 ± 5.7%. The evaluation of patients was made two months after discharge from the hospital, between 10:00 a.m. to 13:00 p.m.

### 2.3. Statistical Analysis

Data are presented as mean ± SD and frequency (%) where appropriate. Data normality was assessed via the Kolmogorov-Smirnov test. Comparisons between sequential continuous data were performed via the paired samples *T*-test. For all tests, a *p*-value < 0.05 was considered statistically significant.

## 3. Results

*Hemodynamic parameters*: Systolic blood pressure in resting (Baseline: 138.7 ± 16.3 vs. post uns-PR: 128.8 ± 8.6 mmHg, t_(19)_ = 3.136, *p* = 0.005, Figure 2) and at the end of 6 MWT (Baseline: 159.8 ± 13.5 vs. post uns-PR: 152.0 ± 12.2 mmHg, t_(19)_ = 2.428, *p* = 0.025, Figure 2) was lower after uns-PR, while didn’t record any difference in diastolic blood pressure before and after uns-PR (*p* > 0.05, Figure 3). Heart rate showed differences in the fifth minute (Baseline: 111.6 ± 16.9 vs. post uns-PR: 105.4 ± 15.9 bpm, t_(19)_ = 2.104, *p* = 0.049, Figure 4) and sixth minute during 6 MWT (Baseline: 112.5 ± 18.3 vs. post uns-PR: 106.9 ± 17.9 bpm, t_(19)_ = 2.218, *p* = 0.039, Figure 4) and as a percent of predicted HR _max_ (Baseline: 70.5 ± 10.9 vs. post uns-PR: 67.0 ± 10.7% of predicted, t_(19)_ = 2.261, *p* = 0.036).

*Oxygen saturation and dyspnea*: Oxygen saturation showed higher values during 6 MWT at the forth minute of 6 MWT (Baseline: 94.6 ± 2.9 vs. post uns-PR: 95.8 ± 3.2%, t_(19)_ = −0.729, *p* = 0.013, Figure 5), first minute of recovery (Baseline: 97.8 ± 0.9 vs. post uns-PR: 97.3 ± 0.9%, t_(19)_ = 2.236, *p* = 0.038, Figure 5) and ΔSpO_2_ (Baseline: 2.8 ± 2.7 vs. post uns-PR: 1.4 ± 2.3%, t_(19)_ = 2.964, *p* = 0.008), compared to baseline values. In dyspnea patients self-assessed lower score at the end of 6 MWT after uns-PR (Baseline: 1.3 ± 1.5 vs. post uns-PR: 0.6 ± 0.9 score, t_(19)_ = 3.135, *p* = 0.005, Figure 6).

*Distance and estimated O_2_ uptake*: Distance showed statistically significant associations before and after 8 weeks uns-PR (Baseline: 433.8 ± 102.2 vs. post uns-PR: 519.2 ± 95.4 m, t_(19)_ = −5.587, *p* < 0.001)and as a percent of predicted values (Baseline: 83.6 ± 17.3 vs. post uns-PR: 99.1 ± 11.4% of predicted, t_(19)_ = −5.971, *p* < 0.001). Leg fatigue before and after uns-PR (*p* > 0.05, Figure 7) was not statistically significant. The estimated O_2_ uptake (Baseline: 14.9 ± 2.4 vs. post uns-PR: 16.9 ± 2.2 mL/min/kg, t_(19)_ = −5.624, *p* < 0.001) and metabolic equivalent (Baseline: 4.3 ± 0.7 vs. post uns-PR: 4.8 ± 0.6 METs, t_(19)_ = −5.514, *p* < 0.001) showed differences before and after unsupervised pulmonary rehabilitation.

*30 s sit to stand**:* Repetitions showed statistically significant associations before and after 8 weeks uns-PR (Baseline: 11.4 ± 3.2 vs. post uns-PR: 14.1 ± 2.7 rep, t_(19)_ = −6.639, *p* < 0.001, Table 3) and ΔSpO_2_ (Baseline: 1.7 ± 1.3 vs. post uns-PR: 1.5 ± 2.3%, t_(19)_ = 2.127, *p* = 0.025, Table 3), compared to baseline values.

*Oxidative stress biomarkers*: Plasma antioxidant capacity showed statistically significant associations before and after 8 weeks uns-PR (Baseline: 2528.3 ± 303.2 vs. post uns-PR: 2864.7 ± 574.8 U.cor., t_(19)_ = −2.401, *p* = 0.027, Figure 8) while reactive oxygen metabolites didn’t show a difference before and after uns-PR significant (Baseline: 335.2 ± 77.1 vs. post uns-PR: 383.2 ± 109.4 U. carr., *p* > 0.05, Figure 9).

*Body composition*: In Table 3 presents the results of patients body composition characteristics between groups before and after 8 weeks uns-PR. Patients after PR observed lower values in parameters percent of body fat (Baseline: 32.2 ± 9.4 vs. post uns-PR: 29.5 ± 8.2%, t_(19)_ = 3.373, *p* = 0.003), visceral fat (Baseline: 14.0 ± 4.4 vs. post uns-PR: 13.3 ± 4.2 score, t_(19)_ = 2.517, *p* = 0.021), neck circumference (Baseline: 39.9 ± 3.4 vs. post uns-PR: 37.8 ± 4.2 cm, t_(19)_ = 3.089, *p* = 0.006) and higher values in parameters muscle mass (Baseline: 30.1 ± 4.6 vs. post uns-PR: 34.6 ± 7.4 kg, t_(19)_ = −2.341, *p* = 0.030) and Δchest (Baseline: 4.2 ± 2.3 vs. post uns-PR: 6.0 ± 1.9%, t_(19)_ = −3.779, *p* = 0.001).

*Sleep quality*: Sleep quality as assessed by PSQI showed differences in before and after PR in parameter “*cannot breathe comfortably*”, (Baseline: 0.6 ± 1.0 vs. post uns-PR: 0.2 ± 0.7, t_(19)_ = 2.333, *p* = 0.031) and “*…..enthusiasm to get things done*” (Baseline: 0.1 ± 0.2 vs. post uns-PR: 0.5 ± 0.6, t_(19)_ = −2.629, *p* = 0.017). Patients after PR decreased the PSQI score (Baseline: 6.7 ± 3.9 vs. post uns-PR: 5.6 ± 3.3, t_(19)_ = 2.258, *p* = 0.036) compared to the period before PR.

## 4. Discussion

Our study indicated that hemodynamic parameters, oxygen saturation, and dyspnea during 6 MWT, in plasma antioxidant capacity, body composition, and sleep quality of participants were among significantly altered indices as a result of uns-PR.

*Differences in SBP and HR during 6 MWT as a result of uns-PR*: A recent study by De Lorenzo et al. [29] reported that COVID-19 survivors presented elevated systolic and diastolic blood pressure at follow-up, requiring pharmaceutical. Fagard et al. [30] reported the combination of aerobic exercise and resistance training decreases blood pressure via the modulation of the sympathetic tone. Collectively, these studies suggest that PR can partially reverse some of the lasting hemodynamic perturbations introduced by COVID-19. Yoga training has been widely used in patients with pulmonary diseases as an alternative exercise program. Positive effects have been reported in various indices, according to meta-analyses [31,32].

*Differences in body composition*: Adipose tissue encompasses several types, with visceral fat being the most clinically relevant as it is associated with more adverse effects compared to peripheral obesity [23]. The interrelationship between obesity and physical activity and cardiovascular diseases is a major underlying mediator of these effects [33]. Moreover, visceral adipose tissue contributes to the activity of a chronic inflammatory substrate [23]. In the present study, were observed reductions in body and visceral fat and increased muscle mass compared to baseline values. In agreement with our findings, Manna and Jain [34] reported on increased muscle activity in overweight patients as a mechanism that induces redox imbalance. Conversely, the Mediterranean diet provides a steady influx of antioxidants that provide a buffer against oxidative stress. These studies could account for the increase in stamina during the 6 MWT observed in our study, reflecting the combined effect of the 8-weeks exercise program Mediterranean diet.

*Oxygen saturation differences during the 6 MWT and 30 Sit to Stand test*: Exercise related changes in oxygenations as a result of uns-PR were observed on both the 30 s sit to stand test and 6 MWT. A previous study [35] has reported that the underlying pathophysiology underlying lasting hypoxia secondary to COVID-19 may be ventilation-perfusion mismatch (VA/Q). Specifically, a combination of intrapulmonary shunting, loss of lung perfusion regulation, and reduced lung compliance may provide the necessary substrate for the latter phenotypical manifestation. A reduction in diffusion capacity is the most commonly reported physiologic impairment [1] due to fibrotic lung remodelling. Exercise widens the alveolar-arterial PO_2_ difference, due to VA/Q, and interstitial pulmonary edema, inadequate ventilatory response, and/or alveolar-capillary diffusion result in further limitation of O_2_ transport [36]. This model could account for the SpO_2_ nadir and HR compensation on the 4th minute of the 6 MWT observed in our study.

Pulmonary rehabilitation programs aim to increase physical activity, improve skeletal muscle function and entrain central desensitization of dyspnea [37]. In our study, patients observed a decrease at the end of 6 MWT in dyspnea, and an increase in the covered distance which led to its increase in estimated O_2_ uptake.

*The effect of uns-PR on sleep quality*: Difficulties in breathing during sleep, such as those reflected by the PSQI item “*cannot breathe comfortably*”, relate to breathing disorders as for example snoring to obstructive sleep apnea [16] while the item *“...enthusiasm to get things done*”, may reflect low motivation due to post-acute COVID-19 syndrome and/or hospitalization in isolation care units [1]. It is generally known that these patients are hospitalized in specialized wards with limited contact with the medical staff [1]. Previous studies have reported on a multifactorial effect of COVID-19 on sleep quality [38]. A survey in a Greek population reported sleep problems prevalent in 37.6% of the participants, during the COVID-19 pandemic [39]. Previous work s from our group [40] indicate the beneficial effect of exercise on sleep quality, even in the setting of sleep disordered breathing.

*The role of a holistic rehabilitation program in the post-COVID-19 setting*: The literature currently lacks data concerning pulmonary rehabilitation programs following COVID-19 recovery. PASC presentations include characterized by dyspnea and myalgia that eventually result in limited exercise capacity [41]. These patients may benefit the most from PR programs and/or telemedicine support [42], however, data on this population are lacking. Our study aimed to provide an in-depth characterization of all the parameters that may be affected by a holistic approach, and those that could be further modified for individualized approaches. Exercise training resulted in an altered O_2_ uptake (functionality index) following an eight-week intervention.

To our knowledge, this is the first study addressing the effectiveness and safety of a pulmonary rehabilitation program in patients with a post-COVID-19. The results of our study may help to establish a specialized rehabilitation program for PASC. Moreover, uns-PR could be a highly valuable tool to promote exercise and symptom recovery following post-COVID-19 as well as a novel approach concerning the treatment of persistently fatigue induced by SARS-CoV-2 infection.

*Limitations, strength, and context:* Our results should be interpreted within our studies’ limitations. Our study’s sample size was negatively influenced by the COVID-19 restriction measures imposed in Greece. Reduced recruitment furthermore led to the lack of positive control for our cohort. A negative control group was not considered, as it would be beyond good clinical practice guidelines. We did not consider multiple comparison adjustments on the premises of the exploratory design of the study [43] and the complementary nature of the tests performed regarding the research question [44], i.e., “what is the physiological effect of uns-PR?”. Our results should be interpreted within this context, and each reported difference should be explored by targeted experiments. Another important limitation is that while statistically significant, specific results may not translate meaningfully in clinical practice. Specifically, BMI and WHR indicate a limited change that neither be communicated to the patient nor represent a robust measure of uns-PR’s biological effect. By contrast, the increase in muscle mass, reduction in neck circumference, and better performance in the 30 s sit-to-stand represent better estimates of the regimen biological effect, that can be easily communicated to patients.

Our study was the first to assess the impact of a fixed training regimen in rehabilitating COVID-19 patients. Improvement was detected on several indices of physical activity, indicating that targeted, holistic rehabilitation may be beneficial in restoring COVID-19 patients’ quality of life.

## 5. Conclusions

We investigated the effects of 8 weeks unsupervised pulmonary rehabilitation program in previously hospitalized COVID-19 patients. We recorded significant alterations in hemodynamic parameters such as oxygen saturation, systolic blood pressure, heart rate, and dyspnea during 6 MWT, in plasma antioxidant capacity, body composition, and sleep quality. According to that, we propose that unsupervised pulmonary rehabilitation could be an effective and beneficial practice to promote exercise and symptom recovery following post-COVID-19 as well as a novel approach concerning the treatment of persistent fatigue induced by SARS-CoV-2 infection.

## Figures and Tables

**Figure 1 jpm-11-00806-f001:**
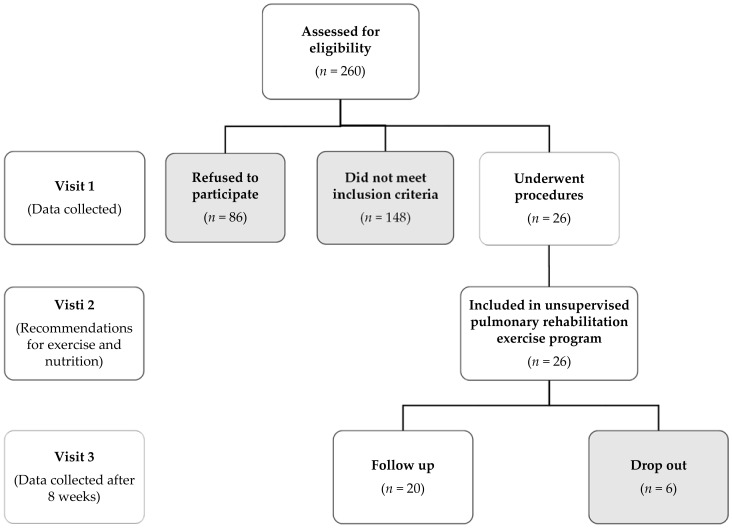
Study flow diagram. Patients which did not meet inclusion criteria were 148. More specifically, 13% with relative contraindications for 6 MWT (e.g., increased resting arterial blood pressure), 22% with inability to complete the balance test, 30% with BMI above 40 kg/m^2^, 8% with mental illness (e.g., depression, anxiety disorders) and 27% with musculoskeletal disability (e.g., injuries of the muscles, joints etc.).

**Figure 2 jpm-11-00806-f002:**
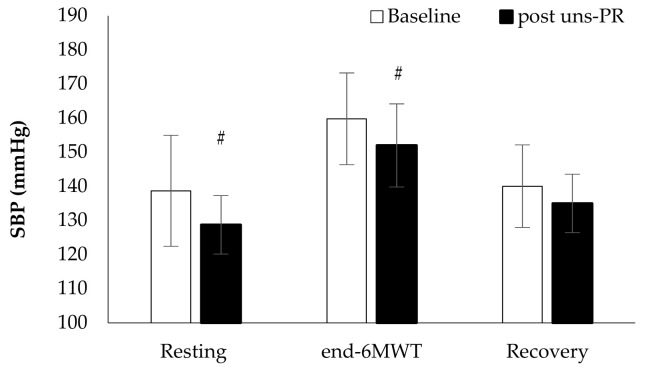
Systolic blood pressure alteration during 6 MWT before and after PR. *# p* < 0.005.

**Figure 3 jpm-11-00806-f003:**
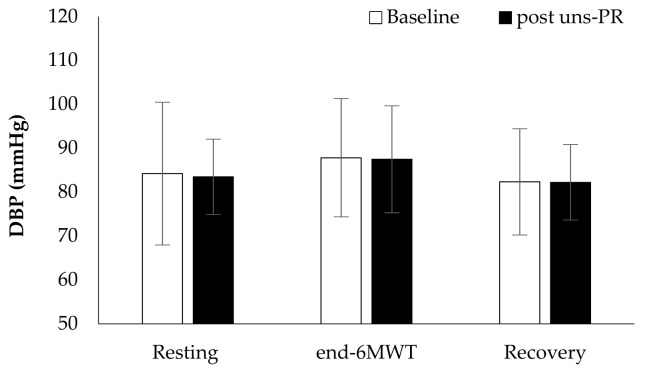
Diastolic blood pressure alteration during 6 MWT before and after PR.

**Figure 4 jpm-11-00806-f004:**
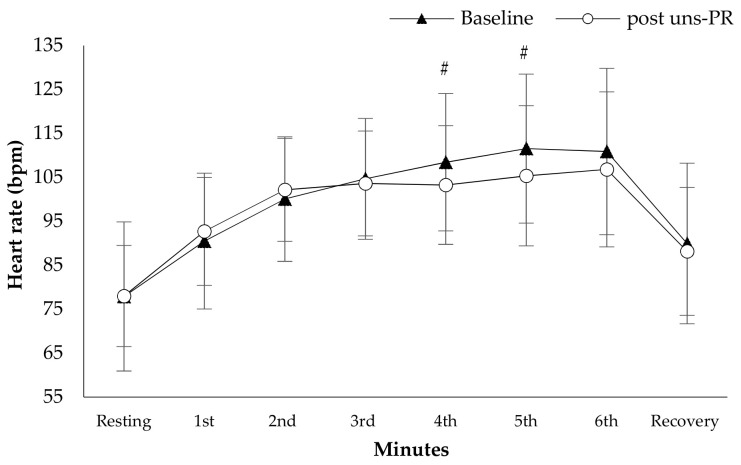
Heart rate alteration during 6 MWT before and after PR. *#*
*p* < 0.05.

**Figure 5 jpm-11-00806-f005:**
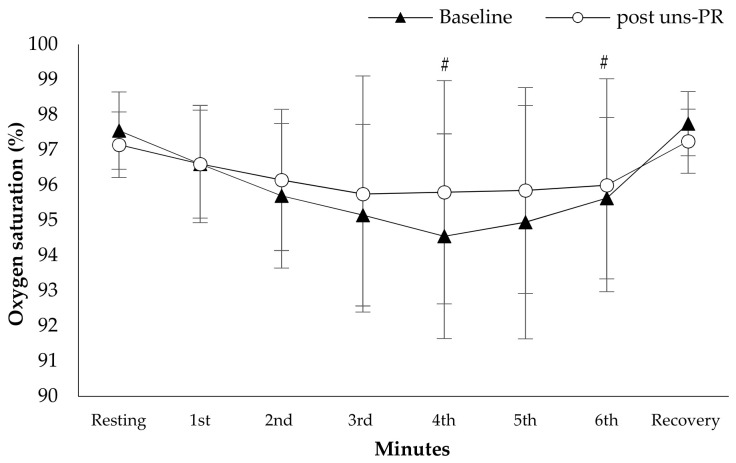
Oxygen saturation alteration during 6 MWT before and after PR. *# p* < 0.05.

**Figure 6 jpm-11-00806-f006:**
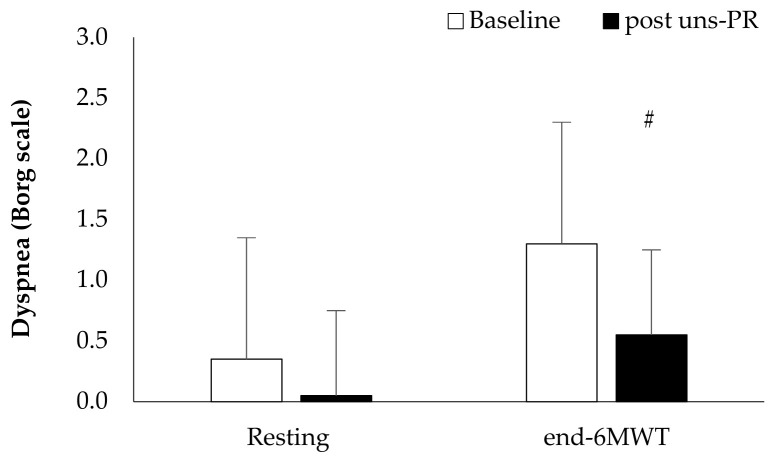
Self-assessed dyspnea pre and post 6 MWT and before and after PR. *# p* < 0.05.

**Figure 7 jpm-11-00806-f007:**
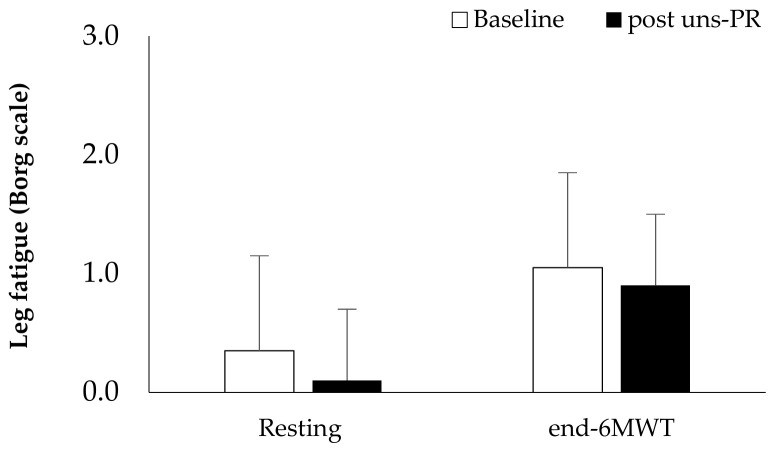
Self-assessed leg fatigue pre and post 6 MWT and before and after PR.

**Figure 8 jpm-11-00806-f008:**
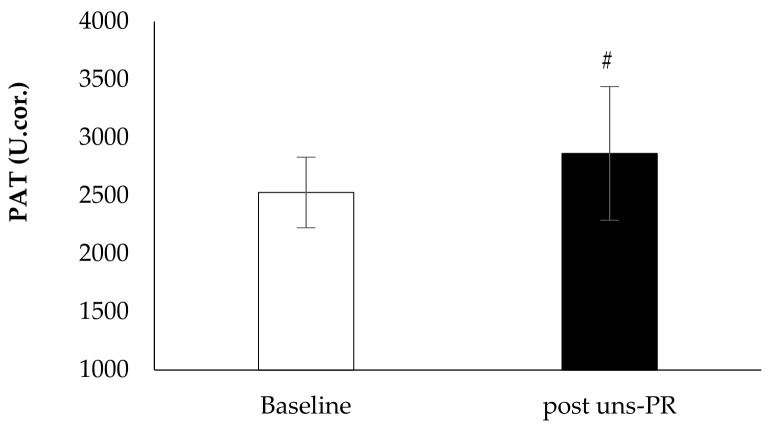
Oxidative stress marker PAT before and after PR. *# p* < 0.05.

**Figure 9 jpm-11-00806-f009:**
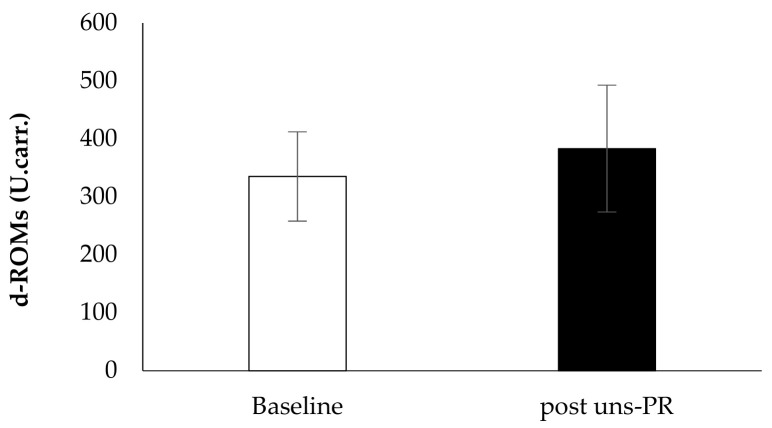
Oxidative stress marker d-ROMs before and after PR.

**Table 1 jpm-11-00806-t001:** Patients demographic characteristics. Data are expressed as mean ± standard deviation or percentages.

	Overall
Age, yrs	64.1 ± 9.9
Gender, (Male %)	75
Smokers, %	10
COPD, %	10
Hypertension, %	65
Diabetes mellitus, %	20
CVD, %	10
Length of hospitalization days	15.1 ± 14.8
Intensive care unit admission, %	20

Abbreviations: COPD = chronic obstructive pulmonary disease; CVD = cardiovascular disease.

**Table 2 jpm-11-00806-t002:** Pulmonary rehabilitation exercise program.

Weeks	Warm-Up/Recovery		Walking		Yoga Breathing		Strength		
	Exercise Type	Sets/ Repetitions (and/or s)	Intensity of HR _peak_ (%)	Borg Scale CR10 (L/D)	Exercise Type	Sets/ Repetitions (and/or s)	Exercise Type	Sets/ Repetitions	Resistance
**1st**	a, b, c	a = 2/6 rep; b = 2/20 s; c = 2/15 s	75	2/3	a, b, c	3/10 rep	a, b	2/12	a = 1.5 kg; b = body weight
**2nd**			80	3/3	a, b, c	4/12 rep	a, b, c	2/12	a = 1.5 kg; b, c = body weight
**3rd**			85	4/4	a, b, c, d	3/12 rep	a, b, c	3/12	a = 1.5 kg; b, c = body weight
**4th**			90	4/5	a, b, c, h	3/12 rep	a, b, c, d	2/12	a = 1.5 kg; b, c, d = body weight
**5th**			95	5/5	e, f, g	3/10 s	a, b, c, d	3/12	a = 1.5 kg; b, c, d = body weight
**6th**			100	5/6	a, b, c, h	4/12	a, b, c, d, e	2/12	a = 1.5 kg; b, c, d = body weight; e = 2 kg
**7th**			105	6/6	a, b, c, h–e, f, g	3/12 rep–4/15 s	a, b, c, d, e	3/12	a = 1.5 kg; b, c, d = body weight; e = 2 kg
**8th**			110	6/6	a, b, c, h–e, f, g	3/12 rep–4/15 s	a, b, c, d, e	3/16	a = 1.5 kg; b, c, d = body weight; e = 2 kg

Abbreviations: D = Borg scale dyspnea; HR _peak_ = heart rate peak during 6-min walk test; L = Borg scale leg fatigue; rep = repetitions; s = sec; Strength exercise = [(a) Dumbbell side lateral raises, (b) Dumbbell squats, (c) Chair lunges, (d) Seated leg raises, (e) Elbow flexion-extension on the chest with medicine ball]; Warm-up/recovery = [(a) Child’s pose—prayer stretch, (b) Doorway stretch, (c) Quadriceps stretch]; Yoga breathing exercise = [(a) Utkatasana, (b) Utthita hasta padangusthasana, (c) Parsvottanasana, (d) Virabhadrasana I, (e) Virabhadrasana II (f) Virabhadrasana III, (g) Vrksasana, (h) Bhujangasa.

**Table 3 jpm-11-00806-t003:** Results analysis before and after unsupervised pulmonary rehabilitation (post uns-PR) in anthropometric and morphologic characteristics, body composition and strength and pulmonary function test parameters. Data are expressed as mean ± standard deviation or percentages.

	Baseline	Post Uns-PR	*p* Value
Body Mass Index, kg/m^2^	30.3 ± 4.3	30.1 ± 4.4	0.030
Body Surface Area, m^2^	2.1 ± 0.3	2.1 ± 0.3	0.110
Body fat, %	32.2 ± 9.4	29.5 ± 8.2	0.003
Visceral fat, score	14.0 ± 4.4	13.3 ± 4.2	0.021
Muscle mass, kg	30.1 ± 4.6	34.6 ± 7.4	0.030
RMR, kcal/day	1725.3 ± 174.7	1739.3 ± 202.9	0.490
Neck circumference, cm	39.9 ± 3.4	37.8 ± 4.2	0.006
WHR	102.6 ± 10.6	101.9 ± 9.4	0.917
Δchest, %	4.2 ± 2.3	6.0 ± 1.9	0.002
Δarm, %	3.7 ± 2.9	3.1 ± 2.2	0.398
Δthigh, %	5.0 ± 4.6	3.4 ± 2.5	0.270
Δcalf, %	3.2 ± 2.8	3.4 ± 3.3	0.809
Handgrip, kg	31.9 ± 10.2	33.2 ± 9.8	0.117
30 s Sit to Stand			
Repetitions/30 s	11.4 ± 3.2	14.1 ± 2.7	<0.001
ΔHR, bpm	23.9 ± 11.2	24.3 ± 15.0	0.903
ΔSpO_2_, %	1.7 ± 1.3	1.5 ± 2.3	0.025
FEV_1_, % of predicted	84.1 ± 18.0	88.2 ± 17.4	0.235
FVC, % of predicted	84.8 ± 15.7	88.6 ± 14.7	0.214
D_LCO_, % of predicted	73.6 ± 11.2	75.7 ± 11.5	0.437

Abbreviations: D_LCO_ = diffusing capacity for carbon monoxide; FEV_1_ = forced expiratory volume in 1st s; FVC = forced vital capacity; RMR = resting metabolic rate; Δarm = arm circumference difference between right and left; Δcalf = calf circumference difference between right and left; Δchest = chest circumference difference between maximal inhalation and exhalation; ΔHR = heart rate difference between resting and post 30 s Sit to Stand test; ΔSpO_2_ = oxygen saturation difference between resting and post 30 s Sit to Stand test; Δthigh = thigh circumference difference between right and left.

## Data Availability

All data are available after request.

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
