# Peer review of "Eight Weeks Unsupervised Pulmonary Rehabilitation in Previously Hospitalized of SARS-CoV-2 Infection"

_jpm, 2021, doi:10.3390/jpm11080806_

Round 1
Reviewer 1 Report
Dear Authors, thank you for submitting your paper.
The aim of the present study was to determine the impact of unsupervised Pulmonary Rehabilitation (uns-PR) on patients recovering from COVID-19, and determine its anthropometric, biological, demographic and fitness correlates
I congratulate the authors for this very relevant research, which will add to the dental field.
It appears well structured, correctly carried out and written without logical or factual errors
Methodological aspects are deeply cleared in the manuscript.
The topic is in line with the journal aim.
-Data reported in the Methods section are appropriate and precisely described;.
-Results are reported clearly and adequately supported by Tables.
-I suggest to the Authors to cite in the Discussion the following recent article about the applications of telemedicine and teleorthodontics during COVID-19 pandemic:
https://doi.org/10.3390/jcm9061891
The Conclusions are correctly stated and supported by the findings obtained from the present study.
According to this Reviewer’s consideration, novelty and quality of the paper, publication of the present manuscript is recommended.
Author Response
Response to reviewers
Paper title: Eight weeks unsupervised pulmonary rehabilitation in previously hospitalized of SARS-CoV-2 infection.
Manuscript ID: jpm-1320458
We thank the reviewers for the comments that have helped us to improve the paper. All changes have been indicated by red color within the text. Below you will find a point-by-point response to your comments.
Reviewer 1
Comments 1. I congratulate the authors for this very relevant research, which will add to the dental field. It appears well structured, correctly carried out and written without logical or factual errors. Methodological aspects are deeply cleared in the manuscript. The topic is in line with the journal aim. Data reported in the Methods section are appropriate and precisely described. Results are reported clearly and adequately supported by Tables. I suggest to the Authors to cite in the Discussion the following recent article about the applications of telemedicine and teleorthodontics during COVID-19 pandemic: https://doi.org/10.3390/jcm9061891 The Conclusions are correctly stated and supported by the findings obtained from the present study.
Response: Thank you for your comment. It has been added the article according suggest of reviewer.
Reviewer 2 Report
The article is very interesting. However, before it could be considered for publication, authors need to incorporate suggestions of the reviewer and extensively revise their manuscript.
General comments Formatting of the contents need to be done as per guidelines of the journal. Sentence formation needs crosscheck. Grammatical mistakes need to be minimized.
Abstract section does not give proper information. Abstract means a full-fledged summary that also give brief highlights of the background information about topics covered in the manuscript. Please revise the abstract.
Section Introduction
Authors are advised to consult following manuscripts on the topic for background information.
Coronavirus Disease -2019 (COVID-19) in 2020: A Perspective Study of a Global Pandemic - PubMed (nih.gov)
A Testimony of the Surgent SARS-CoV-2 in the Immunological Panorama of the Human Host (nih.gov)
Material & Methodology – Study population
Sample number involved in the study is small. Short paras need to be avoided to avoid any confusion between the statements. I think that reason for not including control can be removed.
Figures
Major problem with figures is that their error bars are too large. It seems data not statistically significant.
Discussion
It is too long. Authors need to present it in a more concise manner and highlight the significance of their study in the light of recent reports about COVID.
Section conclusion
The section should highlights importance of the study and future directions with possible limitations.
Author Response
Response to reviewers
Paper title: Eight weeks unsupervised pulmonary rehabilitation in previously hospitalized of SARS-CoV-2 infection.
Manuscript ID: jpm-1320458
We thank the reviewers for the comments that have helped us to improve the paper. All changes have been indicated by red color within the text. Below you will find a point-by-point response to your comments.
Reviewer 2
General comments: Formatting of the contents need to be done as per guidelines of the journal. Sentence formation needs crosscheck. Grammatical mistakes need to be minimized.
Comments 1. Abstract section does not give proper information. Abstract means a full-fledged summary that also give brief highlights of the background information about topics covered in the manuscript. Please revise the abstract.
Response: Thank you for your comment. It has been revised the abstract according suggest of reviewer.
Comments 2. Section Introduction: Authors are advised to consult following manuscripts on the topic for background information. Coronavirus Disease -2019 (COVID-19) in 2020: A Perspective Study of a Global Pandemic - PubMed (nih.gov). A Testimony of the Surgent SARS-CoV-2 in the Immunological Panorama of the Human Host (nih.gov).
Response: Thank you for your comment. We have read very carefully the manuscripts and we obtained relevant information.
Comments 3. Material & Methodology – Study population: Sample number involved in the study is small. Short paras need to be avoided to avoid any confusion between the statements. I think that reason for not including control can be removed.
Response: Thank you for your comment. We cannot remove the reason for not including control group because the method should be consistent with the approve by the IRB/EC of the University Hospital of Larissa.
Comments 4. Figures: Major problem with figures is that their error bars are too large. It seems data not statistically significant.
Response: Thank you for your comment. In figures the statistical significance presented with symbol and not in all parameters
Comments 5. Discussion: It is too long. Authors need to present it in a more concise manner and highlight the significance of their study in the light of recent reports about COVID.
Response: Thank you for your comment. The segment of discussion it has been re-written according suggest of reviewer.
Comments 6. Section conclusion: The section should highlights importance of the study and future directions with possible limitations.
Response: Thank you for your comment. It has been added the section “Limitations, strength, and context”.
Reviewer 3 Report
The submitted work aimed to determine the impact of unsupervised PR (uns-PR) on patients recovering from COVID-19, and determine it's anthropometric, biological, demographic and fitness correlates. Several improvements were found after an 8-week intervention, indicating that uns-PR might be effective. The authors are congratulated for including a comprehensive assessment of included participants and putting the study together during strict regulations. However, I do have some concerns.
- First of all, although motivated in the materials and methods section, the lack of a control group is a significant methodological limitation of the submitted work. Without a control group, it is difficult to establish if the observed effects would be due to the uns-PR program provided or simply due to normal recovery? Therefore, the lack of a control group and its potential impact on the study findings need to be further discussed. Could comparisons be made with previous work that ha followed post-COVID individuals over time?
- Although significant improvements were seen on several outcomes, numerically, the changes seem to be relatively small. Therefore, I would here recommend discussing/highlight not only if the findings are statistically significant but also if they may be clinically relevant?
- Another related limitation is that the results were analyzed per protocol. Thus, only 20 of the 26 participants included in the study was part of the primary analysis (30% drop-out rate). I would recommend that the authors perform an intention-to-treat analysis and not a per-protocol analysis.
- Additionally, many comparisons are made; however, as far as I can tell, no adjustments to the statistical analysis have been performed, correcting for multiple comparisons. Such a correction is also recommended.
- It is concluded that the intervention is safe. However, no safety measures/objectives seem to be included in the trial?. On what do the authors base this assumption?
- The selected intervention is described as unsupervised pulmonary rehabilitation (uns-PR). However, the selected exercises do not seem to fit the recommendations for pulmonary rehabilitation programs. E.g., the use of Yoga breathing exercises and the selection of included resistance exercises. In general, more detailed information on the training program used is necessary, specifically if the selected intervention would be implemented in a clinical setting and to facilitate replication of the trial. See the TIDIER guidelines for reporting interventions: BMJ 2014;348:g1687 doi: 10.1136/bmj.g1687
- As seen in Figure 1, only 10% of the potentially eligible participants participated in the study, reducing the generalizability of the findings. The authors are encouraged to discuss this further. On the ame topic, if possible, adding more information on the 90% of people assessed for eligibility that either refused to participate or did not meet inclusion criteria would be meaningful to understand for whom uns-PR might be relevant. E.g., was any of the selected exclusion criteria more often reported than the other? Was it the more severe individuals who declined to participate etc?
Minor comments
- Would suggest adding supporting data (not only p-values) in the abstract.
- Page 2, lines 62-64. Add supporting references to the statements made in these sentences that are important as these provide the rationale for the study.
- Page 4, line 136. When were the assessments performed? Add this information, if it was completed the first or the last day of hospital admission, is essential information for interpreting the study's findings. At the moment, it is only stated that the evaluation was made between 10:00-13.00.
- The results section is somewhat difficult to read. Since all data (?) is provided in the attached tables and figures, I would strongly recommend reducing the amount of information provided in the text.
Author Response
Response to reviewers
Paper title: Eight weeks unsupervised pulmonary rehabilitation in previously hospitalized of SARS-CoV-2 infection.
Manuscript ID: jpm-1320458
We thank the reviewers for the comments that have helped us to improve the paper. All changes have been indicated by red color within the text. Below you will find a point-by-point response to your comments.
Reviewer 3
The submitted work aimed to determine the impact of unsupervised PR (uns-PR) on patients recovering from COVID-19, and determine it's anthropometric, biological, demographic and fitness correlates. Several improvements were found after an 8-week intervention, indicating that uns-PR might be effective. The authors are congratulated for including a comprehensive assessment of included participants and putting the study together during strict regulations. However, I do have some concerns.
Comments 1. First of all, although motivated in the materials and methods section, the lack of a control group is a significant methodological limitation of the submitted work. Without a control group, it is difficult to establish if the observed effects would be due to the uns-PR program provided or simply due to normal recovery? Therefore, the lack of a control group and its potential impact on the study findings need to be further discussed. Could comparisons be made with previous work that ha followed post-COVID individuals over time?
Response: Thank you for your comment. It was difficult to comparison our patients with subjects from previous works because there aren't corresponding studies and enough method information with similar exercise program.
Comments 2. Although significant improvements were seen on several outcomes, numerically, the changes seem to be relatively small. Therefore, I would here recommend discussing/highlight not only if the findings are statistically significant but also if they may be clinically relevant?
Response: Thank you for your comment. It has been added information in section “Limitations, strength, and context”.
Comments 3. Another related limitation is that the results were analyzed per protocol. Thus, only 20 of the 26 participants included in the study was part of the primary analysis (30% drop-out rate). I would recommend that the authors perform an intention-to-treat analysis and not a per-protocol analysis.
Response: Thank you for your comment. It has been added information in section “Limitations, strength, and context”.
Comments 4. Additionally, many comparisons are made; however, as far as I can tell, no adjustments to the statistical analysis have been performed, correcting for multiple comparisons. Such a correction is also recommended.
Response: Thank you for your comment. The lack of multiple comparisons, it has been added in section “Limitations, strength, and context”
Comments 5. It is concluded that the intervention is safe. However, no safety measures/objectives seem to be included in the trial?. On what do the authors base this assumption?
Response: Thank you for your comment. It has been replaced “safe exercise” with “beneficial practice”.
Comments 6. The selected intervention is described as unsupervised pulmonary rehabilitation (uns-PR). However, the selected exercises do not seem to fit the recommendations for pulmonary rehabilitation programs. E.g., the use of Yoga breathing exercises and the selection of included resistance exercises. In general, more detailed information on the training program used is necessary, specifically if the selected intervention would be implemented in a clinical setting and to facilitate replication of the trial. See the TIDIER guidelines for reporting interventions: BMJ 2014;348:g1687 doi: 10.1136/bmj.g1687
Response: Thank you for your comment. The study of Hoffmann et al. it was very helpful. It has been re-written the section of pulmonary rehabilitation exercise program.
Comments 7. As seen in Figure 1, only 10% of the potentially eligible participants participated in the study, reducing the generalizability of the findings. The authors are encouraged to discuss this further. On the ame topic, if possible, adding more information on the 90% of people assessed for eligibility that either refused to participate or did not meet inclusion criteria would be meaningful to understand for whom uns-PR might be relevant. E.g., was any of the selected exclusion criteria more often reported than the other? Was it the more severe individuals who declined to participate etc?
Response: Thank you for your comment. It has been added information about the patients which did not meet inclusion criteria, in study flow diagram.
Minor comments
Comments 8. Would suggest adding supporting data (not only p-values) in the abstract.
Response: Thank you for your comment. It has been revised the abstract according suggest of reviewer.
Comments 9. Page 2, lines 62-64. Add supporting references to the statements made in these sentences that are important as these provide the rationale for the study.
Response: Thank you for your comment. It has been added reference.
Comments 10. Page 4, line 136. When were the assessments performed? Add this information, if it was completed the first or the last day of hospital admission, is essential information for interpreting the study's findings. At the moment, it is only stated that the evaluation was made between 10:00-13.00.
Response: Thank you for your comment. It has been added “The evaluation of patients was made two months after discharge from hospital between 10:00 a.m. to 13:00 p.m”.
Comments 11. The results section is somewhat difficult to read. Since all data (?) is provided in the attached tables and figures, I would strongly recommend reducing the amount of information provided in the text.
Response: Thank you for your comment. It has been reduced the amount of information provided in the text.
Round 2
Reviewer 2 Report
The authors have successfully answers all my queries and the MS can be accepted for publication......
Author Response
General comment: The authors have successfully answers all my queries and the MS can be accepted for publication......
Response: We thank the reviewer for this comment.
Reviewer 3 Report
Thank you for the revised version of your manuscript. Several improvements to the submitted work have been made, but some need to be further addressed. The original comments are listed below.
COMMENT 1: Although motivated in the materials and methods section, the lack of a control group is a significant methodological limitation of the submitted work. Without a control group, it is difficult to establish if the observed effects would be due to the uns-PR program provided or simply due to normal recovery?
The point-by-point response stated that it was “difficult to compare our patients with subjects from previous works because there aren't corresponding studies and enough method information with similar exercise program”.
The comment was a bit misinterpreted. What was meant is that comparisons could be made with studies that follow COVID patients over time and see how the “normal recovery” is regarding the selected outcomes. No PR program would be needed. However, there are studies available that have used PR in post-COVID patients that the results of the present trial (even if the PR intervention is not identical)) e.g., https://pubmed.ncbi.nlm.nih.gov/33800094/ https://pubmed.ncbi.nlm.nih.gov/33588090/
Since the current study lacks a control group, putting the findings of the studies in relation to normal recovery and/or to other PR programs used on post-COVID is essential to interpret the magnitude of effect in this exploratory trial.
COMMENT 3. Another related limitation is that the results were analyzed per protocol. Thus, only 20 of the 26 participants included in the study was part of the primary analysis (30% drop-out rate). Therefore, I would recommend that the authors perform an intention-to-treat analysis and not a per-protocol analysis.
The point-by-point response stated that “information had been added in the “Limitations, strength, and context” section”. However, the recommendation was to make an ITT analysis, not to keep the per-protocol. Keeping only the per-protocol analysis can lead to erroneous conclusions and increases the risk for bias. Another suggestion would be to present the results of both an ITT and a per-protocol.
COMMENT 6. If the TIDIER checklist were used, I would recommend adding a reference highlighting this. When examining the new Table 2, the uns-PR program is much easier to interpret. Great additions!
I am still a bit curious about the Yoga breathing exercises and the strength exercises. The former usually are not part of PR programs, and the latter does not seem to be executed as recommended in PR programs. E.g., using weights corresponding to at least 60% of 1RM for strength exercises as highlighted in reference 9 in the introduction.
Therefore, I would recommend addressing the setup of the uns-PR program in the discussion, highlighting that it is not performed following general PR guidelines.
Author Response
COMMENT 1: Although motivated in the materials and methods section, the lack of a control group is a significant methodological limitation of the submitted work. Without a control group, it is difficult to establish if the observed effects would be due to the uns-PR program provided or simply due to normal recovery? The point-by-point response stated that it was “difficult to compare our patients with subjects from previous works because there aren't corresponding studies and enough method information with similar exercise program”. The comment was a bit misinterpreted. What was meant is that comparisons could be made with studies that follow COVID patients over time and see how the “normal recovery” is regarding the selected outcomes. No PR program would be needed. However, there are studies available that have used PR in post-COVID patients that the results of the present trial (even if the PR intervention is not identical)) e.g., https://pubmed.ncbi.nlm.nih.gov/33800094/ https://pubmed.ncbi.nlm.nih.gov/33588090/ Since the current study lacks a control group, putting the findings of the studies in relation to normal recovery and/or to other PR programs used on post-COVID is essential to interpret the magnitude of effect in this exploratory trial.
Response: We thank the reviewer for this comment. Our study presents in section "Limitations, strength, and context" the lack of control group and the small sample size. It is difficult to redesign of the study.
COMMENT 3. Another related limitation is that the results were analyzed per protocol. Thus, only 20 of the 26 participants included in the study was part of the primary analysis (30% drop-out rate). Therefore, I would recommend that the authors perform an intention-to-treat analysis and not a per-protocol analysis. The point-by-point response stated that “information had been added in the “Limitations, strength, and context” section”. However, the recommendation was to make an ITT analysis, not to keep the per-protocol. Keeping only the per-protocol analysis can lead to erroneous conclusions and increases the risk for bias. Another suggestion would be to present the results of both an ITT and a per-protocol.
Response: We thank the reviewer for this comment. While 6 participants did drop out and represent a limitation, intention-to-treat analysis cannot be performed within the context of our study and its design. There are specific reasons for this:
- Our study did not include a control arm.
- Our study is single arm (i.e. everyone was treated, and hence there is no "per-protocol")
- There was no randomization process prior to patient allocation to the program.
- There were no binary or categorical (in general) outcomes, and hence, there is no way to attribute drop-out patients to alternative hypotheses (i.e. as these are not tested)
- Our study is exploratory, i.e. it does not have a predicted outcome, binary or otherwise, that is strictly and specifically expected to change. Instead, the research question is broadly attributed to "How does this specific uns-PR regimen affect fitness in these patients?"
- Simulation approaches that would allocate extrapolated values (i.e. by imputing their values as if they were "unchanged", rejecting the null hypothesis that uns-PR would result in significant changes) would be the only way to practically perform ITT in a relevant way. However, this is (a) not performed in exploratory studies where data on an intervention is either scarce or absent and (b) has a questionable biological basis, since this approach would have to construct virtual cases, effectively feeding from the samples' variance but not contributing to the overall information drawn from the sample.
We agree that in the case of a study that would include at least two intervention arms and binary, predetermined outcomes (i.e. death, survival, or any other milestone) ITT could minimize bias introduced by drop-out or non participation. As our study is not structured as a randomized case-control trial, and the "treatment" condition applies to our (single-arm) population, per-protocol is the only feasible approach.
COMMENT 6. If the TIDIER checklist were used, I would recommend adding a reference highlighting this. When examining the new Table 2, the uns-PR program is much easier to interpret. Great additions! I am still a bit curious about the Yoga breathing exercises and the strength exercises. The former usually are not part of PR programs, and the latter does not seem to be executed as recommended in PR programs. E.g., using weights corresponding to at least 60% of 1RM for strength exercises as highlighted in reference 9 in the introduction. Therefore, I would recommend addressing the setup of the uns-PR program in the discussion, highlighting that it is not performed following general PR guidelines.
Response: We thank the reviewer for this comment. Pulmonary rehabilitation is a program of education and exercise to increase awareness about your lungs and your disease. (https://www.lung.org/lung-health-diseases/lung-procedures-and-tests/pulmonary-rehab). Our uns-PR program, does not follow general the PR guidelines and exercise prescription practices in pulmonary rehabilitation programs with traditional concept for two reasons: First of all the there are no PR program and/or guidelines, except the study of Spruit et al. with holistic recommendation for COVID-19 as an interim guidance on rehabilitation in the hospital and post-hospital phase from a ERS and ATS coordinated International Task Force, instructions based on COPD PR programs. Secondly, the unsupervised PR it not so famous due to risk wrong execution of exercise which can lead to injury. The usp-PR programs it was a necessary choice due to there are objective socio-economic adversities and difficulties in transfer of patients in PR centers and re-infection due to co-morbidity.